# Betaxolol as a Potent Inhibitor of NDM-1-Positive *E. coli* That Synergistically Enhances the Anti-Inflammatory Effect in Combination with Meropenem

**DOI:** 10.3390/ijms241713399

**Published:** 2023-08-29

**Authors:** Jichao Sun, Shangjie Ren, Yaozu Yang, Xiaoting Li, Xiuying Zhang

**Affiliations:** 1Heilongjiang Key Laboratory for Animal Disease Control and Pharmaceutical Development, Northeast Agricultural University, Harbin 150030, China; caassjc@163.com (J.S.); renshangjie2022@163.com (S.R.); yangyaozu2022@163.com (Y.Y.); lixting3069@163.com (X.L.); 2Department of Basic Veterinary Science, College of Veterinary Medicine, Northeast Agricultural University, Harbin 150030, China

**Keywords:** betaxolol, repositioning, New Delhi metallo-β-lactamase, inhibitor discovery

## Abstract

With significant human and economic losses, increasing bacterial resistance is a serious global threat to human life. Due to their high efficacy, broad spectrum, and cost-effectiveness, beta-lactams are widely used in the clinical management of bacterial infection. The emergence and wide spread of New Delhi metallo-β-lactamase (NDM-1), which can effectively inactivate β-lactams, has posed a challenge in the design of effective new antimicrobial treatments. Medicine repurposing is now an important tool in the development of new alternative medicines. We present a known glaucoma therapeutic, betaxolol (BET), which with a 50% inhibitory concentration (IC_50_) of 19.3 ± 0.9 μM significantly inhibits the hydrolytic activity of the NDM-1 enzyme and may represent a potential NDM-1 enzyme inhibitor. BET combined with meropenem (MEM) showed bactericidal synergism in vitro. The efficacy of BET was further evaluated against systemic bacterial infections in BALB/c mice. The results showed that BET+MEM decreased the numbers of leukocytes and inflammatory factors in peripheral blood, as well as the organ bacterial load and pathological damage. Molecular docking and kinetic simulations showed that BET can form hydrogen bonds and hydrophobic interactions directly with key amino acid residues in the NDM-1 active site. Thus, we demonstrated that BET inhibited NDM-1 by competitively binding to it and that it can be developed in combination with MEM as a new therapy for the management of infections caused by medicine-resistant bacteria.

## 1. Introduction

β-lactam antibiotics are some of the most important antibiotics and are widely used in medical clinics and agricultural farming due to their broad antimicrobial spectrum and high efficacy [1,2]. The unreasonable use of β-lactam antibiotics leads to survival pressure on bacteria, producing β-lactamases (BLs) that hydrolyze the nucleus of the β-lactam ring, making Gram-negative bacteria that are resistant to carbapenems and most other β-lactam antibiotics. This poses a serious challenge to antimicrobial therapy, and the resulting antimicrobial medicine resistance (AMR) poses a major threat to public health worldwide [3,4].

The β-lactamase superfamily contains thousands of enzymes that vary in their genes and functions. Due to resistance to almost all known β-lactam antibiotics, New Delhi metallo-β-lactamase-1 (NDM-1) has attracted much attention in clinical and community-based medicine-resistant bacterial infections. NDM-1, as one of the most hydrolytically efficient enzymes among the metallo-β-lactamases (MBLs) [5] of subclass B1, can hydrolyze the β-lactam ring via a nucleophilic attack on the β-lactam carbonyl group using two Zn^2+^-bound hydroxyl groups [6], thereby creating resistance to β-lactams. NDM-1 has become endemic in more than 70 countries worldwide. Due to the localization of the gene in the plasmid, it is highly metastatic in a broad spectrum of clinically important Gram-negative bacteria, including *E. coli* and *Acinetobacter baumannii*. NDM-1 often comes with other resistance genes, making it multi-drug-resistant (MDR) to almost all types of antibiotics and causing infections with mortality rates of up to 51% [7]. An effective way to overcome β-lactamase resistance is to combine antibiotics with enzyme inhibitors, but the active-site compositions and surroundings of BLs show a high degree of variability across subclasses, making the development of new inhibitors difficult. Currently, the only inhibitors approved by the FDA for clinical use to inhibit the development of serine β-lactamases (SBLs) are clavulanic acid, sulbactam, tazobactam, and the recently marketed avibactam, and vaborbactam, while inhibitors targeting NDM-1 have yet to be consistently explored [8].

For inhibitor discovery, the current approach is often through time-consuming and labor-intensive high-throughput screening. However, as protein structures are being resolved, medicine screening can be performed by targeting the active sites of bis-zinc ions through the computer-assisted molecular simulation of docking, and particular attention is paid to the repositioning of FDA-approved medicines during the screening process, which can help to reduce the time and cost of medicine development.

Through preliminary screening, we obtained betaxolol [9,10], which can inhibit NDM-1, and it showed inhibitory efficacy against the NDM-1 enzyme in vitro. We tested the antimicrobial activity of BET+MEM in vitro using a checkerboard assay, temporal bactericidal curves, and an interventional cellular infection assay and further investigated the in vivo efficacy of BET+MEM in infected mice. The competitive inhibition of NDM-1 active-site activity by BET was determined via molecular docking and kinetic simulation studies. In conclusion, BET could be repositioned or developed as an inhibitor of NDM-1.

## 2. Results

### 2.1. BET Has Enzymatic Inhibitory Activity against NDM-1

The drug BET, which has been approved by the FDA for clinical use in combination with our engineered NDM-1 enzyme expression system, has been repositioned as an NDM-1 inhibitor. We assessed the ability of recombinant NDM-1 to hydrolyze the substrate in the presence of BET using a nitrocefin assay. The inhibition ratio (IR) of BET at a concentration of 10 μg/mL against NDM-1 could reach 87.5%, and the 50% inhibitory concentration (IC_50_) was 19.3 ± 0.9 μM, which is comparable to the inhibitory activity of D-mercaptopurine (7.9 μM) and showed significant dose dependence (Figure 1).

### 2.2. MIC and FICI of BET+MEM against NDM-1 E. coli

BET’s efficacy in inhibiting natural NDM-1 was tested using clinical strains of porcine-derived *E. coli* that can produce NDM-1. When testing the minimal inhibitory concentration (MIC) value for NDM-1 *E. coli*, the MEM value was interpreted as 64 μg/mL, and the BET result was well over 1024 μg/mL (Table 1). The inhibitory concentration (FIC) was evaluated using the EUCAST-guided checkerboard method. When BET was co-administered with MEM, BET (128 μg/mL) reduced the MIC of MEM from 64 μg/mL to 8 μg/mL, and the inhibitory concentration index (FICI) was 0.25, indicating that the two have a synergistic antimicrobial effect.

### 2.3. Growth Effect of NDM-1 E. coli Treated with BET+MEM

The growth curve of NDM-1 *E. coli* was not significantly affected by BET alone (in the concentration range of 0–128 μg/mL). In the concentration range of 0–128 μg/mL, BET had no significant effect on the growth of the test strains, but BET (128 μg/mL) restored the inhibitory effect of MEM (8 μg/mL) on the growth of the bacteria, which indicated that BET (128 μg/mL) was not detrimental to the growth of the NDM-1 *E. coli* but restored the performance of MEM (Figure 2).

### 2.4. Potential Toxicity of BET to IECs

To investigate the potential cellular toxicity of BET, we used CCK-8 to determine the damage of different concentrations of BET co-incubated with IECs for 24 h. As shown in Figure 3, the cellular activity remained above 95% for 24 h of exposure to BET concentrations in the range of 0–400 μM, which did not induce necrosis of IECs.

### 2.5. Intervention of IECs with BET+MEM in NDM-1 E. coli Infection

To investigate the performance of BET and MEM in NDM-1 *E. coli*-infected IECs, BET alone or in combination with MEM was used to intervene in NDM-1 *E. coli*-infected mouse intestinal epithelial cells (moi = 100). As illustrated in Figure 4, compared to the model (DM) group, the BET+MEM group reduced the bacterial adhesion to the cells with 60% inhibition, and the LDH concentration in the cell culture solution was significantly reduced (*p* < 0.5), indicating that the combination of BET and MEM reduced the adhesion of NDM-1-positive *E. coli* to the IECs and attenuated the damage of the IECs (Figure 4).

### 2.6. BET+MEM Treatment of NDM-1 E. coli-Infected Mice

Using in vitro and cellular studies, we demonstrated that BET reversed the susceptibility of NDM-1 bacteria to MEM and that BET+MEM could destroy NDM-1 *E. coli*. In order to further clarify the potential of BET, we performed an in vivo assay to verify whether BET could show results in vivo that corresponded to the in vitro assay. BALB/c mice received intraperitoneal injections of NDM-1 *E. coli* (1 × 10^7^ CFU) and were treated in different ways. Only 20% of the mice in the model (DM) and BET groups survived. After treatment with MEM, the survival rate increased to 40%. After the BET+MEM treatment, 70% of the mice were still alive after 48 h, which was significantly higher than in the DM group (*p* < 0.05). The combination of BET+MEM significantly increased the survival rate of the infected mice (Figure 5).

The BET+MEM combination increased the survival of mice infected with a lethal dose of NDM-1 *E. coli*, so the effect of the BET+MEM combination on the mouse model of bacterial inflammation was further explored. Groups were treated after an intraperitoneal infection with NDM-1 *E. coli* (1 × 10^6^ CFU). Leukocytes and serum TNF-α, IL-1β, and IFN-γ were measured in peripheral blood after the euthanasia of the mice 96 h after infection (Figure 6A–D), and it was found that the indexes of the inflammatory factors were higher in the DM group, the BET group, and the MEM group than in the MEM+BET group, and the indexes in the DM group and the BET group were higher than those in the MEM group (*p* < 0.05). The bacterial counts in the liver, spleen, and intestine were significantly lower in the BET+MEM group than in the DM, BET, and MEM groups (Figure 6E).

A histopathological (H&E staining) examination revealed that, as shown in Figure 6F, compared with the DM group, after MEM alone or BET+MEM treatment, the hepatocytes in the liver were regularly and densely arranged, with the central vein as the center in a cord-like arrangement in all directions. Additionally, the boundaries between the white medulla and red medulla regions of the spleen were more clearly defined, the cell shapes were rounded or ovoid, and the intestinal enterochromaffin structures were nearly clear and intact, with only a small number of necrotic cells (black arrows). After BET alone, similar to the DM group, the liver was infiltrated with a large number of neutrophils, some nuclei disappeared or lysed, and the hepatocytes were degenerated and necrotic and the necrotic hepatocytes were fused into sheets. In the spleen, there was blurring of the boundaries between the white and red medullary areas, irregular cell shape and increased interstitial space, congestion of and reduction in lymphocytes in the red medullary area, almost complete disintegration of the intestinal villous structure, necrotic detachment of epithelial cells, and a significant reduction in the number of cup-shaped cells (red arrows). Taken together, the above results showed that the combined treatment of BET+MEM could effectively reduce the inflammatory response of infected mice.

In conclusion, the combination of BET and MEM showed synergistic antimicrobial effects in vivo; reduced mortality in NDM-1 *E. coli*-infected mice; significantly lowered the level of inflammation in infected mice and the number of bacteria in the target organs; and attenuated pathological damage in the liver, spleen, and intestine.

### 2.7. Molecular Docking and Kinetic Simulation Analysis

To further explore possible binding patterns of BET to NDM-1 proteins, molecular docking with NDM-1 proteins using BET was carried out, and docking calculations showed that BET formed close interactions with NDM-1 active sites, amino groups, and hydroxyl groups (Figure 7A,B). The hydroxyl group interacted with the backbone amide group of Glu152 to form hydrogen bonds, and the secondary amino group formed an attractive charge with the backbone amide group of the Glu152 and Asp223 side chains. The cyclopropane group formed alkyl bonds with His250, Ile35, and Lys211, while the ether group interacted with Lys211.

### 2.8. MM/GBSA Combined Free-Energy Calculations and Decompositions

To explore possible binding patterns and observe changes in the conformation of the complex, BET was attached to NDM-1’s active site, the BET-NDM-1 system was constructed, and molecular dynamics (MD) simulations were performed for a duration of 150 ns. The root-mean-square deviation (RMSD) value increased rapidly from 0 to 12 s and stabilized after 22 ns, the displacement of the main chain atoms in the whole MD simulation stage was small, and the composite system quickly reached a state of dynamic equilibrium, indicating that the conformation of the complex system could be stable in the solvent state (Figure 8).

The free-energy results of the MM/GBSA binding of proteins to small-molecule compounds in the BET-NDM-1 complex system are shown in Table 2. The free energy of the total binding (ΔG_bind_) of the BET-NDM-1 system was -11.32 ± 4.62 kcal/mol, and the interaction between BET and NDM-1 was strong. Although the electrostatic force (ΔE_ele_) in the system was large, the overall binding free energy of the system was mainly due to the ΔE_vdw_, which was due to the hydrophobic interactions between the molecules, and the polar solvent effect (ΔG_GB_) almost completely counteracted the electrostatic attraction force. When considering the solvent effect, the total electrostatic effect (ΔE_vdw_+ΔG_GB_) contributed little or even negatively to the interaction between systems.

The free-energy binding of each residue pair in the NDM-1 protein to BET was quantitatively calculated using the free-energy decomposition calculation method to study the total free-energy binding of the complex system of the NDM-1 and BET complex. The free energy of the binding of MM/GBSA was broken down, and the amino acid residues that contributed more to the free energy of the binding were labeled (Figure 9).

In order to understand the interaction of the complex system more intuitively, after decomposing the MM/GBSA binding free energy, the binding mode of the MD simulation system was analyzed at 150 ns. After 150 ns of MD simulation, the active hydrolysis center of NDM-1 was stably bound by BET. The MD simulation results showed that during the binding and interaction between BET and NDM-1, most of the energy came from the various interaction forces formed by NDM-1 and that the inhibitor BET hydrolyzed the central amino acid residues (His122, Asp124, Glu152, His189, and His250). It also formed an electrostatic force with metal Zn^2+^ at the active site of the NDM-1 enzyme, stably binding to the active hydrolysis center of NDM-1, so the activity of the NDM-1 enzyme was reduced. Thus, the combination of BET and NDM-1 may inhibit the activity of the NDM-1 enzyme through competitive inhibition.

## 3. Discussion

β-lactam antibiotics occupy an important position in global antimicrobial medicine by inhibiting bacterial cell-wall mucopeptide synthetases (penicillin-binding proteins, PBPs), thus hindering the synthesis of cell-wall mucopeptides, which results in bacterial cell-wall defects, leading to bacterial bloom, lysis, and death.

However, recent studies with large amounts of resistance-monitoring data have shown that the massive expression of MBLs by Enterobacteriaceae bacteria and their delivery in bacteria has become a global public health problem [11,12]. In particular, *E. coli* containing the BL enzyme have become a thorny problem in the clinical treatment of *E. coli* infections due to its stable and effective hydrolysis of almost all β-lactam antibiotics, coupled with the uncontrollable nature of clinical dosing. To date, inhibitors of BLs more specific to SBLs, such as tazobactam, clavulanate [13], and sulbactam [14], are FDA-approved and have been a significant improvement compared to penicillin and cephalosporin antibiotics in terms of antibacterial activity alone. However, although there are no approved NDM-1 inhibitors, some NDM-1 inhibitors reported in recent articles, such as vidofludimus [15], baicalin [16], isoliquiritin [17], etc., have certain inhibitory activity. In light of the difficulties in developing new antibiotics and the urgency of medicine resistance, the development of innovative alternative treatment strategies for NDM-1 has become critical.

It is generally accepted that NDM-1 is one of the main reasons for bacterial resistance to β-lactams. BET exhibited good inhibition of NDM-1 (IR of 87.5%, IC_50_ of 19.3 ± 0.9 μM) and can be developed as a potential NDM-1 inhibitor. BET, as a selective β-adrenergic receptor antagonist, was once the first-line treatment for open-angle glaucoma [18,19]. It can also be used to treat high blood pressure [20,21,22].

The results of the checkerboard test showed that NDM-1 *E. coli* was resistant to MEM (64 μg/mL). The addition of BET reversed this phenomenon and restored MEM resistance, and they synergistically killed NDM-1 *E. coli.* Based on this, our laboratory is trying to use NDM-1 inhibitors in clinical practice and develop BET as a new NDM-1 inhibitor.

Thus, using growth curves, time–kill curves, and an IEC model of *E. coli* infection, we evaluated the antimicrobial activity of BET+MEM in vitro. The results showed that the combination of BET (128 μg/mL) and MEM (8 μg/mL) exhibited a significant synergistic bactericidal effect at 0–5 h, which may have been due to the addition of BET and MEM in the early stage of bacterial proliferation (an important period for cell-membrane and cell-wall formation), and the slight rebound of the growth curves in the later stage may have been the result of the depletion of BET (compared to the MEM group). At the same time, the combination reduced the adhesion of bacteria to the cells and their ability to infect them.

However, due to the different internal physiological environments of animals, there may be differences in their abilities to absorb and metabolize different drugs, so in vitro synergism does not accurately reflect in vivo synergism. Therefore, model animals, BALB/c mice, were injected intraperitoneally with NDM-1 *E. coli*, and it was further evaluated whether BET could restore the susceptibility of NDM-1 *E. coli* to MEM in vivo. Since NDM-1 *E. coli* carries a variety of virulence factors, such as bacterial toxins and adhesins, after successfully infecting the mice at a dose of 1 × 10^7^ CFU, the mortality rate at 120 h was 80%. We determined that BET (10 mg/kg) + MEM (10 mg/kg) was effective in reducing mortality, the peripheral blood leukocyte count, inflammatory factor levels, the organ bacterial load, and histopathological damage in infected mice through mouse survival and treatment experiments. These results suggest that BET restores MEM sensitivity in vivo.

In previous reports, BET has been used as a selective β-1 blocker for the treatment of glaucoma and hypertensive eye disease by inhibiting calcium influx and in young hypertensive patients with moderate hypertension or without serious complications.

We also demonstrated that 400 μg/mL BET showed no signs of toxicity to IECs, which was consistent with the previously reported safe dose of 40 mg/kg. In practice, BET is usually administered via ocular perfusion or orally. Long-term toxicity tests have demonstrated that BET (48 mg/kg in rats and 60 mg/kg in mice) is not teratogenic, mutagenic, or carcinogenic.

Although most of the current reports on NDM-1 inhibitors are in the discovery or preclinical development stages, more than 500 compounds with good activity are available for further development. For example, captopril, a well-known drug for the treatment of hypertension, chelates the metal zinc ions of NDM-1. D-captopril (IC_50_ of 7.9 μM) and L-captopril (202.0 μM) have been repositioned as potential NDM-1 inhibitors. Magnolia powder, a natural compound isolated from the bark of Magnolia officinalis, is a natural product with a similar mechanism. It exhibits a highly selective inhibition of NDM-1 and restores the antibacterial activity of MEM against *E. coli*.

MD is now a routine technique for studying the dynamic motions of biological macromolecules at the atomic resolution level, providing an immediate, cross-affinity, and intuitively interactive visual understanding of complex dynamic trajectories, and in 2013, the Nobel Prize in Chemistry was awarded to Martin Karplus in recognition of the relevance and importance of MD from the perspective of developing multiscale models of complex chemical systems. This study also explored the binding mechanism of BET to NDM-1 using molecular docking and simulated kinetics.

## 4. Materials and Methods

### 4.1. Strains, Cells, and Culture Conditions

Porcine-derived *E. coli* was isolated and identified as a strain capable of producing NDM-1 by our laboratory [23]. *E. coli* BL21(DE3)-pET-32a(+)-blaNDM-1 was constructed and preserved by our laboratory [15], and *E. coli* ATCC25922, as a quality control strain, was purchased from the Chinese Centre of Medical Strain Preservation and Management (Beijing, China). These strains were cultured in Luria–Bertani medium (LB) and shaken overnight at 37 °C. Intestinal epithelial cells (IECs, purchased from the Shanghai Institute of Cell Biology, Chinese Academy of Sciences, Shanghai, China) were cultured in RPMI-1640 complete medium containing 10% FBS and 1% penicillin/streptomycin at 37 °C, 5% CO_2_, and 70–80% humidity.

### 4.2. Chemicals

Betaxolol (BET, purity ≥ 98%) and meropenem (MEM, purity ≥ 98%) were purchased from Target Mol (Boston, MA, USA) and dissolved in dimethyl sulfoxide to make a stock solution that was stored at −80 °C.

### 4.3. Nitrocefin Assay and IC_50_ Assay

As the β-lactam ring of nitrocefin (NIT) was opened by the hydrolysis of the NDM-1 enzyme, the substrate changed from yellow to red, which was used to determine the inhibitory effect of BET on the hydrolytic activity of NDM-1. The NDM-1 enzyme was prepared and purified using *E. coli* BL21(DE3)-pET-32a(+)-blaNDM-1 as described in a previous article [15]. After incubating the NDM-1 enzyme with different concentrations of BET (0–128 μg/mL) in HEPES buffer (containing 50 μM ZnSO_4_, pH = 7.5) at 37 °C for 5 min, 100 μL of NIT was added as a substrate. After 10 min of incubation, the hydrolysis level of NIT was determined at OD_492_ nm. The IC_50_ value was defined as the concentration of the inhibitor that would inhibit 50% of the enzyme [24].

### 4.4. In Vitro Bacteriostatic Activity Determination

The EUCAST guid recommended the broth microdilution method, which was used to determine the MIC values of BET, MEM, and BET+MEM against NDM-1 *E. coli*. The fractional inhibitory concentration (FIC) was assessed using the EUCAST-guided tessellated grid assay. An FIC of ≤0.5 was defined as synergistic [25].

### 4.5. Bacterial Growth and Sterilization Curves

Purified NDM-1 *E. coli* was inoculated into 200 mL of an LB broth medium at 1:1000 and cultured until OD_600 nm_ = 0.3. The bacterial solution was added to media containing 0, 8, 16, 32, 64, and 128 μg/mL BET and incubated for 6 h at 37 °C and 200 rpm with oscillation, and the OD_600 nm_ was detected and recorded every 1 h. The growth curve was plotted to determine the influence of BET on the growth of NDM-1 *E. coli*.

The synergistic antibacterial power of BET+MEM on NDM-1 *E. coli* was evaluated as described by Meher Rizvi [26].

### 4.6. CCK-8 Assay

The CCK-8 assay was used for the determination of the potential toxicity of BET to mammalian cells. IECs (1 × 10^4^ cells/well) were inoculated into 96-well plates and cultured overnight (37 °C, 5% CO_2_, 70–80% humidity) to 80% fusion [27]. The cells were grown in an antibiotic-free RPMI-1640 medium containing 1% FBS, and the BET concentration in the wells was adjusted. After 24 h, a CCK-8 solution was added and incubated for 1 h at 37 °C in the dark. A Multiskan SkyHigh Full Wavelength Enzyme Labeller (Thermo Fisher Scientific Inc., Waltham, MA 02451, USA) was used to measure the absorbance at OD_450 nm_.

### 4.7. LDH Assay and Bacterial Count

An LDH assay was used to assess whether BET reversed the susceptibility of MEM to NDM-1 *E. coli* infection in IECs. IECs were cultured in 96-well plates until fusion. Then, the medium was discarded and washed with PBS. Next, 100 μL of RPMI-1640 medium containing 1 × 10^5^ CFU of NDM-1 *E. coli* was added. After incubation for 2 h at 37 °C and 5% CO_2_, one part was added to an LDH detection reagent (Beyotime Biotech. Inc., Shanghai, China) and incubated for 30 min at 37 °C in the dark to compare the absorbance at 490 nm with the untreated group. The other portion was washed with sterile 1 × pbs (pH 7.0) and serially diluted and counted after overnight incubation at 37 °C on antibiotic-containing plates to assess the effect of BET+MEM on NDM-1 *E. coli*-infected IECs [28].

### 4.8. Animal Research

Female BALB/c mice weighing 18.0–22.0 g (Animal License No.: SCXK(Liao)2020-0001) were purchased from Liaoning Changsheng Laboratory Animal Co. The mice were kept in an SPF animal house (25 ± 2 °C, 50% relative humidity, 12:12 h light–dark cycle) for 1 week to adapt to the experimental environment. All experiments were reviewed and approved by the Animal Protection and Utilisation Committee of Northeast Agricultural University (APUC number: NEAUEC20230399). All experimental methods, animal care, and in vivo experiments in this study were conducted in strict accordance with the animal ethical procedures of the People’s Republic of China.

NDM-1 *E. coli* (lethal dose, 1 × 10^7^ CFU) was administered to BALB/c mice, and the survival rate of BET combined with MEM was investigated in NDM-1 *E. coli*-infected mice. Except for the blank control group (NC), the mice were randomly divided into 4 groups after bacterial infection, i.e., the infection model group (DM), the BET treatment group (BET), the MEM treatment group (MEM), and the BET combined with MEM treatment group (BET+MEM), with 10 mice in each group. The medicines were administered 2 h after infection and once every 12 h for a total of 6 times. After 24, 48, 72, 96, and 120 h, the number of surviving mice was calculated [29,30,31].

To evaluate the protective effect of BET+MEM against NDM-1 *E. coli* infection, mice were injected intraperitoneally with NDM-1 E. coli (1 × 10^6^ CFU). The grouping and dosing methods were the same as those used in the survival assay. Whole blood was collected after 96 h to determine the leukocyte content, and serum was used to determine the levels of the inflammatory factors TNF-α, IL-1β, and IFN-γ in serum were detected by ELISA kits (Beijing Chenglin Biotechnology Co., Ltd., Beijing, China). After aseptic separation of the liver, spleen, and intestine, a certain weight of the organs was weighed and homogenized with an automatic sample quick grinder at 12,000 r/m for 2 min, serially diluted, and spread on LB agar plates (containing MEM) to count the number of bacteria after overnight incubation at 37 °C. And the other part was fixed in 10% formalin and stained with hematoxylin and eosin H&E for histopathological examination.

### 4.9. Molecular Docking and MD Simulation

In order to clarify the binding interactions between NDM-1 and BET, molecular docking analysis was performed using Discovery Studio 2016 (BIOVIA Corp, San Diego, CA, USA). The crystal structure of NDM-1 bound to hydrolyzed methicillin (ID: 4EY2, resolution: 1.17 Å, http://www.rcsb.org, assessed on 7 February 2022) was selected as the receptor. The chemical structure of BET was obtained from the PubChem website (https://pubchem.ncbi.nlm.nih.gov, assessed on 7 February 2022) and as the chemical structure of the ligand compounds for an intuitive molecular docking analysis [32,33].

Simulated kinetic analysis was performed using AMBER 20 (University of California, San Francisco, CA, USA), and the optimal complex for molecular docking, BET-NDM-1, was selected as the starting conformation for the MD study. In the CHARM36 force field, the system was added in the vicinity of TIP3P water molecules, and the BET-NDM-1 complex was determined to extend outward by 12 Å as the spatial extent of the complexes in the simulation, with the addition of Na^+^ and K^+^ ions to make up for the lack of charge in the force field, and was optimized for energy minimization using the sander program. After a 30 ns warm-up, 120 ns MD simulations were performed on a stable system at a constant temperature and pressure of 300 K and 1 atmosphere, retaining the conformational trajectory of the system every 2 ps. MM/GBSA free-energy analyses, MD trajectory information calculations, and kinetic interaction analyses were performed for BET and NDM-1 amino acids.

## 5. Conclusions

In conclusion, BET was repositioned as an anti-NDM-1 inhibitor. BET showed direct interactions with key active-site amino acids in NDM-1 to restore the susceptibility of NDM-1 *E. coli* to MEM by suppressing its hydrolytic activity against β-lactam antibiotics. In vivo, the combined application of BET and MEM (10 mg/kg + 10 mg/kg) significantly increased the survival of NDM-1 *E. coli*-infected mice. Significant changes in peripheral blood leukocyte counts, inflammatory factor levels, and bacterial loads in organs, as well as the histopathology in the mice, indicated that BET restored the susceptibility of NDM-1 *E. coli* to MEM and attenuated inflammatory responses in vivo. The combination of BET and MEM provides a new therapeutic strategy against β-lactam-resistant bacterial infections. This finding will help to utilize existing approved drugs by screening for new targets and repositioning them as inhibitors of NDM-1 or even MBL, providing research ideas for the treatment of infections with “superbug” pathogens.

## Figures and Tables

**Figure 1 ijms-24-13399-f001:**
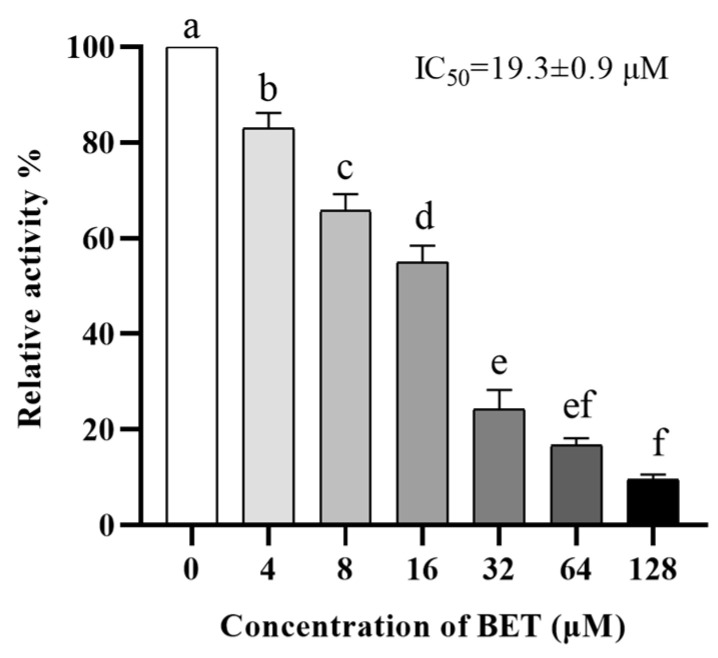
The ability of betaxolol (BET) to inhibit New Delhi metallo-β-lactamase (NDM-1) enzyme. Three independent replicates for each concentration. Significant differences are indicated by different lower-case letters (*p* < 0.05).

**Figure 2 ijms-24-13399-f002:**
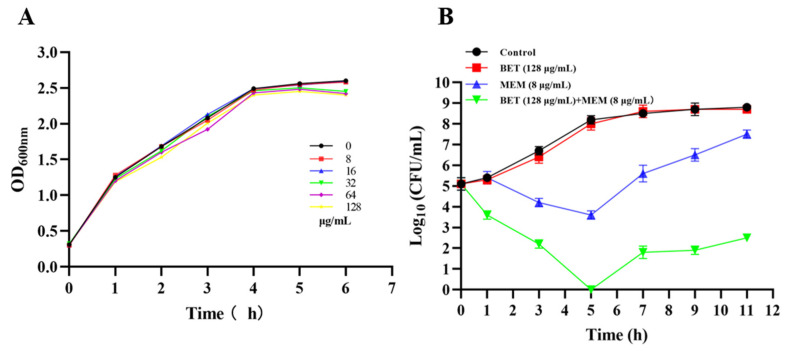
BET restores susceptibility of NDM-1 *E. coli* to MEM. (**A**) Growth of NDM-1 *E. coli* had no effect from BET (0–128 μg/mL). (**B**) BET restored the susceptibility of NDM-1 *E. coli* to MEM compared to MEM application alone. Each experiment was repeated three times independently.

**Figure 3 ijms-24-13399-f003:**
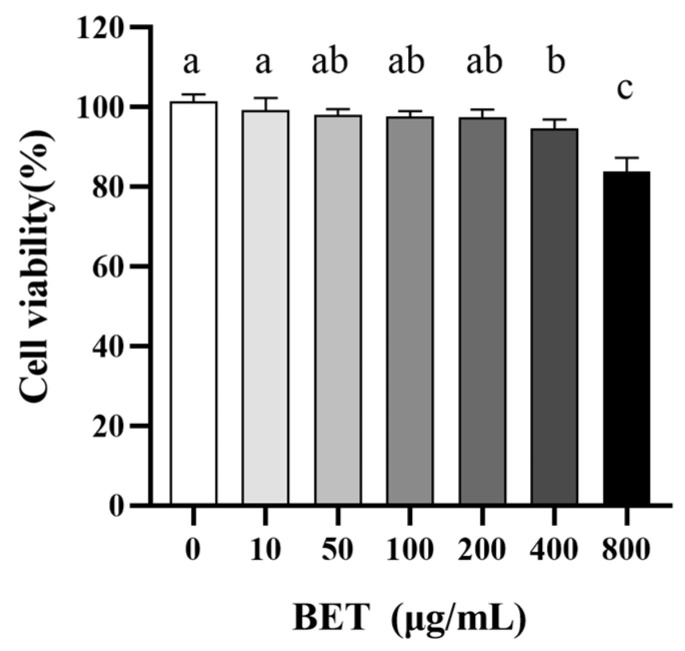
Toxicity of BET to IECs. Significant differences are indicated by different lower-case letters (*p* < 0.05).

**Figure 4 ijms-24-13399-f004:**
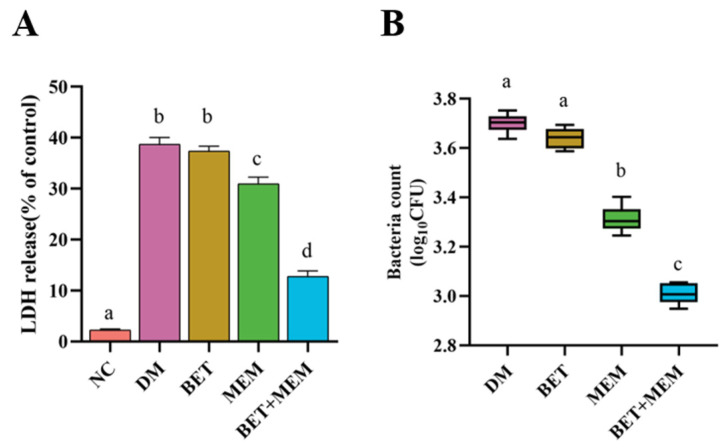
The combination of BET and MEM reduced the ability of bacteria to infect (**A**) and adhere to IECs (**B**). Significant differences are indicated by different lower-case letters (*p* < 0.05).

**Figure 5 ijms-24-13399-f005:**
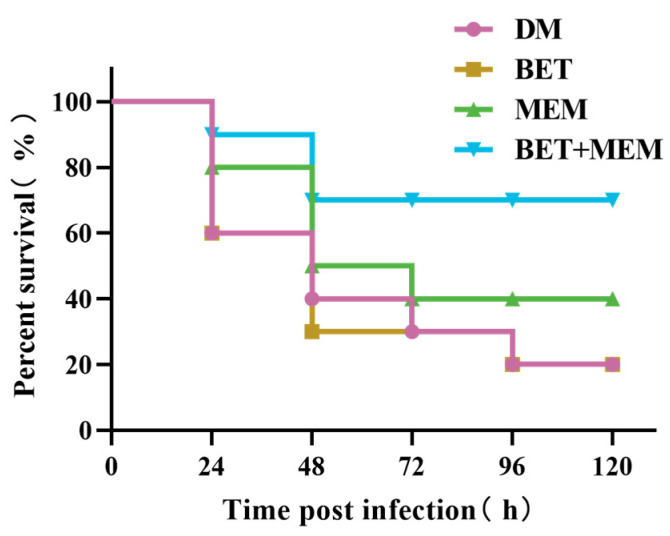
Survival of infected mice was improved by BET combined with MEM. BALB/c mice were injected intraperitoneally with NDM-1 *E. coli* (lethal dose, 1 × 10^7^ CFU) and subsequently divided into four groups: DM group, MEM group (10 mg/kg), BET group (10 mg/kg), and BET+MEM group (10 mg/kg + 10 mg/kg).

**Figure 6 ijms-24-13399-f006:**
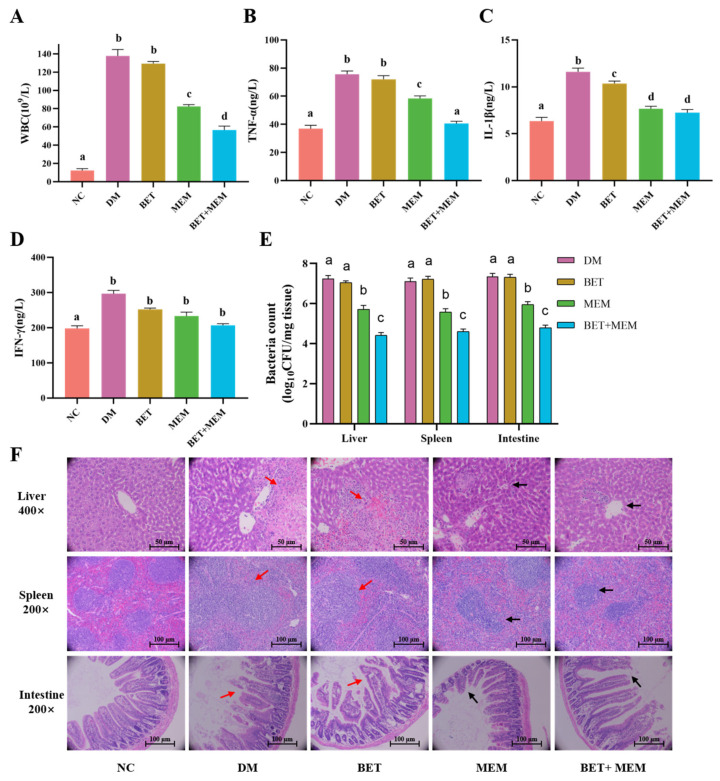
Effects of different treatment groups in vivo. NDM-1 *E. coli* (1 × 10^6^ CFU) was injected intraperitoneally into BALB/c mice. They were divided into the NC group, DM group, BET group (10 mg/kg), MEM group (10 mg/kg), and BET+MEM group (10 mg/kg + 10 mg/kg). (**A**) Number of leukocytes in different groups of mice (n = 10). (**B**) TNF-α levels in different groups (n = 10). (**C**) IL-1β levels in different groups (n = 10). (**D**) IFN-γ levels in different groups of mice (n = 10). (**E**) Liver, spleen, and intestinal bacterial loads in different groups of mice (n = 10). (**F**) Histopathological changes in liver, spleen, and intestines of mice in different groups. Significant differences are indicated by different lower-case letters (*p* < 0.05).

**Figure 7 ijms-24-13399-f007:**
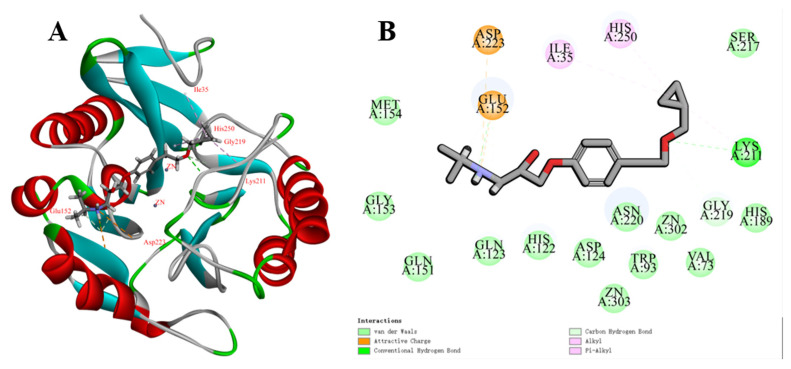
Molecular structure of BET and its docking complex structure and mode of action with NDM-1 molecule. (**A**) Structure of BET and NDM-1 molecular docking. The red area in Figure 7 indicates the α-helical structure of the protein, and the blue area indicates the β-sheet structure of the protein. (**B**) BET and NDM-1 molecular docking mode 2D analysis.

**Figure 8 ijms-24-13399-f008:**
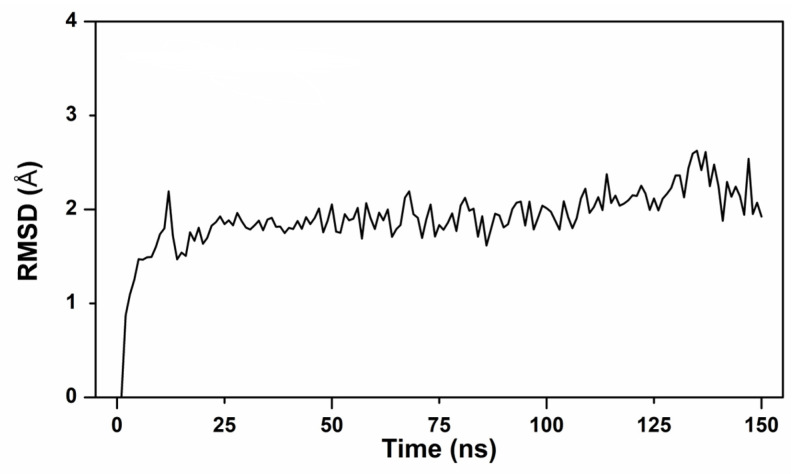
RMSD changed with MD duration.

**Figure 9 ijms-24-13399-f009:**
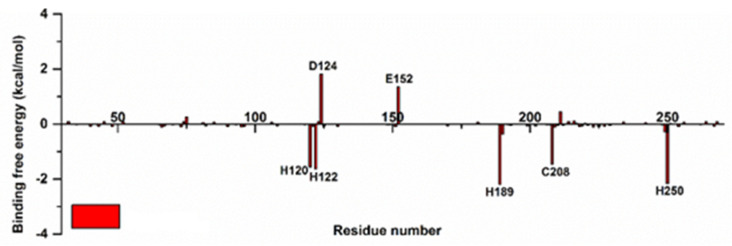
The free energy of binding for each amino acid residue in the system.

**Table 1 ijms-24-13399-t001:** MIC and FICI of NDM-1 *E. coli* for BET+MEM.

Medicine	MIC (μg·mL^−1^)		FICI	Interaction Mode
	Applied Separately	Applied Combined		
MEM	64	8	0.25	Synergistic
BET	1024	128		

**Table 2 ijms-24-13399-t002:** Calculation of the MM/GBSA binding free energy of the system (kcal/mol).

System	ΔE_vdw_	ΔE_ele_	ΔG_GB_	ΔG_SA_	ΔG_bind_
BET-NDM-1	−12.88 ± 3.63	−107.33 ± 23.04	110.64 ± 14.21	−1.74 ± 0.69	−11.32 ± 4.62

## Data Availability

Not applicable.

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
