# Peer review of "Betaxolol as a Potent Inhibitor of NDM-1-Positive E. coli That Synergistically Enhances the Anti-Inflammatory Effect in Combination with Meropenem"

_ijms, 2023, doi:10.3390/ijms241713399_

Round 1

Reviewer 1 Report

The article entitled ”Betaxolol as a potent inhibitor of NDM-1 that synergistically enhances the anti-inflammatory effect in combination with meropenem” provides an interesting insight into repurposing old medication. It is a well written article with comprehensive evidence. However, I will address my concerns as follows:

·        The authors tested in vivo activity of these drugs only in infections produced by NDM1-positive E. coli. I strongly advice to include in the title this aspect to not create confusion. Although preliminary data support their findings, since it was tested only for E. coli there are no evidence to present these results as a general finding.

·        Line 14 – bacterial inflammation. Do you mean infection?

·        Line 50 – multi medicine resistant (MDR) – perhaps rephrase to multi drug resistant (MDR)

·        Line 122 – infection ability of E. coli. Not in general.

·        Figure 8  - explain in the legend of that figure the meaning of RMSD.

·        4.4. – Why do you used two standards? CLSI and EUCAST? Please explain.

Reviewer 2 Report

The manuscript entitled “Betaxolol as a potent inhibitor of NDM-1 that synergistically enhances the anti-inflammatory effect in combination with meropenem”, submitted for evaluation to IJMS, presents the evaluation of betaxolol effect on NDM-1 enzyme activity in vitro and in vivo, in the enzyme-containing bacteria.

In general, the work is clear and written in the correct language. The structure of the manuscript is relatively clear. Combination of in silico and experimental results brings significant novelty to state-of-the-art in the area of reversion of bacterial resistance to known antibiotics (in relation to beta-lactams and New Delhi metallo-beta-lactamanse activity).

To increase the quality of this manuscript, I submit some comments and proposals:

 COMMENTS TO AUTHORS

 1.  Please write the names of bacteria and (in vitro) and (in vivo) in italics.

2.  Section 2.5.: It is well known that LDH is the marker of cell damage and toxicity in reaction to different agents. Please explain on what basis the authors assumed that the measurement of LDH can be a marker of bacterial adhesion to IECs? Please also explain in the "methods" section the procedure for the determination of this marker in co-cultures and IECs and provide the sources of the kits used for LDH determination.

3. Section 2.6.  Please describe (and mark in Figure 6F) what changes indicate the histopathological changes that the authors refer to, explaining that BET reduces pathological changes in tissues (so that the conclusions are clear to a non-expert).

4. Figure 7: what blue and red regions indicate?

5. Page 9, lines 266-267: Note: “all the mice died within 120 h…”  seems to be false because in Figure 5 it was shown that in all groups at least 20% of all animals survived after 120 h.

6. Section 4.8.: Please provide the number of APUC agreement. What are the doses of both drugs applied to mice? Please provide the source of reagents (kits) used for determination of inflammatory factors. How bacterial content in collected organs was evaluated (please describe the method)?

No significant language corrections required.

Author Response

请参阅附件。

Reviewer 3 Report

Title: Betaxolol as a potent inhibitor of NDM-1 that synergistically enhances the anti-inflammatory effect in combination with meropenem

The study has a sound methodology.

The analysis has been described well. 

The discussion is clear and balanced.

I have a few minor comments:

Line 49              'E. coli and Acinetobacter baumannii'. Italicise the genus and                             species name throughout the manuscript.

Line 50                Do the authors mean Multi-Drug Resistant?

Figures                Authors should specify what a, b, c, e and f stand for in the                              legend.

 Which statistical analysis method was used?
